# Early Detection of Phenotypic Diversity of Alfalfa (*Medicago sativa* L.) in Response to Temperature

**DOI:** 10.3390/plants12183224

**Published:** 2023-09-11

**Authors:** Abraham J. Escobar-Gutiérrez, Lina Q. Ahmed

**Affiliations:** INRAE, UR4 P3F, Le Chêne—BP 6, F-86600 Lusignan, France; ahmedlin.inra16@gmail.com

**Keywords:** breeding, germination, heterotrophic growth, hypocotyl, legumes, radicle

## Abstract

Climate change may have important consequences on plant distribution because local environments could change faster than the pace of natural selection and adaptation of wild populations and cultivars of perennial forages. Temperature is a primary factor affecting seed germination and primary heterotrophic growth processes. *Medicago sativa* (L.) is the most important forage legumes globally. The accelerated breeding of alfalfa cultivars adapted to new ranges of temperature could be necessary under most future climate scenarios. This work aims to explore the genetic diversity of a sample of accessions for responses to temperature during seed germination and seedling heterotrophic growth. Seeds or seedlings were placed in the dark under eight constant temperatures in the range of 5 °C to 40 °C. Germinated seeds were manually counted, while hypocotyl and radicle growth were estimated by using image analysis and curve fitting. Multivariate analyses highlighted links between responses and the origin of accessions. Variability was high, within and between accessions, for all the response variables. Accessions showed significant differences in their non-linear response curves in terms of germinability, germination rates and relative elongation rates. Nevertheless, differences were more noticeable in germination rations and rates compared to seedling heterotrophic growth. Consequently, these could be easier to use as early markers for alfalfa selection and breeding for the future.

## 1. Introduction

Climate change may have important consequences on plant distribution in the near future. Indeed, ongoing global climate change is affecting local annual courses of precipitation and temperatures. Further, it is expected to become increasingly severe in temperate areas. Consequently, it will certainly cause a rising number of warm summers and will short the growing season with important consequences on plant distribution and ecosystems reshaping. It might result in both an increase in unsuitable areas for cropping and degraded yields across the globe [1].

Recent studies suggest that local environmental changes are faster than the pace of natural selection and adaptation of wild populations of perennial plants in natural grasslands. This is true for perennial grasses [2,3] and legumes [4] species cultivated in temporary grasslands. Thus, climate change is becoming a major threat to wild and cultivated grasslands. Furthermore, the performances of the current panel of commercial cultivars could also be affected. Any adaptation of cultivated plants in response to temperature and precipitation changes may considerably influencecrop functioning and performance [5].

One of the current challenges of forage breeders is to create new cultivars adapted to future climate conditions. To this end, exploiting the genetic diversity found in wild populations could be of paramount importance for breeding [2,3,6]. Additionally, high-density genomic data could help identify valuable germplasm and be used in the development of selection tools for breeding [7]. However, the genomic bases of environmental adaptation remain poorly understood [2,3,8].

Temperature is a primary climate factor controlling and regulating plant development (i.e., plant phenology, organogenesis and expansive growth) due to its effects on metabolic and physiological rates [9]. For example, seed germination [10] and heterotrophic growth [11] rates respond to temperature in a form that is coherent with theory [12]. Indeed, germination is susceptible to environmental cues that can influence individual fitness, population persistence, and the spatial distribution of species [13,14].

Heterotrophic growth, happening in the dark, as well as germination, is under complex control. These two processes depend on the multiple interactions of an individual’s genotype (endogenous), the individual’s trait established or acquired during seed development (e.g., seed physiological quality), and environmental conditions (exogenous) [15]. The growth of seedling parts, such as radicle, epicotyl, coleoptile or hypocotyl, according to species, plays an essential role in the emergence process for the reason that it leads the seedlings outside of the soil. It is mainly based on both (i) the relative growth rate and (ii) the final length that this axis can reach, which are vigor traits correlated with emergence [16,17,18].

Alfalfa (*Medicago sativa* L.) is the most important and oldest cultivated forage crop in the world [6,19,20]. This perennial forage species is of great interest because of its nitrogen-fixing capability, which is important in the present phase of agroecological transition, and its use in ruminant feeding [21]. Modern *M. sativa* is a complex of eight diploid or auto-tetraploid perennial and allogamous subspecies from different geographical zones with contrasting annual temporal ambient temperatures. Cultivated alfalfa is an auto-tetraploid plant showing a higher vigor than diploid subspecies. Most European varieties have undergone introgression from subsp. *falcata*, which provides cold tolerance and allows the breeding of varieties for Northern areas [6,19,20]. The accelerated breeding of alfalfa cultivars adapted to new ranges of temperature could be necessary under most future climate scenarios. To such an end, we have previously explored and analyzed the diversity of responses to temperature during the germination of accessions of alfalfa [6,22]. However, in addition to germination, the initial heterotrophic growth of seedlings is crucial to cover establishment. Thus, our objective for this study was to analyze a sample of accessions of alfalfa for their response to constant temperature during primary heterotrophic growth. A subsidiary effort focused on establishing relationships between germination and heterotrophic growth rates. These might be important early phenotypic markers in breeding programs.

## 2. Results

### 2.1. Germination

The extent of genetic variation in seed germination in response to temperature of seven accessions of alfalfa previously published [22] was assessed using multivariate techniques. Principal component analysis of germination data is summarized in Table 1, Figure 1 and Figure 2.

Six components accounted for 100%. Four components explained 91.7% of the total variability contained within the 24 germination-related variables included in the analysis (Table 1). Furthermore, the first two components accounted for 72.4% of the variability. This analysis allowed a practical condensation of the data (Figure 1). Indeed, it was observed that the eight variables of maximum germinability, *G_max_T_*, contributed each with more than 5% (Figure 2A) to component 1. The major contributors to component 2 were variables related to the germination lag, *t_c_T_*, under the various treatments (Figure 2B). The ten major contributors to component 3 were a mix of variables of germinability, *G_max_T_*, germination rates, *α__T_*, and germination lag, *t_c_T_*, at different temperatures (Figure 2C).

Hierarchical clustering on principal components (HCPC) by alfalfa accession in response to temperature during germination produced the results presented in Figure 3. Accessions were positioned according to their contribution to components 1 and 2. Three groups can be distinguished by their location in the plot. The first group is composed of the four varieties of subsp. *sativa*. The second group includes the landrace ‘Flamande’ and the variety of subsp. *falcata* ‘Krasnokutskaya’. The distance of these two accessions on component 2 is large but they are well aligned on component 1. Far from them, on the axis of component 1, is the other landrace, ‘Demnate3’.

### 2.2. Initial Heterotrophic Growth

We used fifteen sequential photographs per seedling to follow up, measure and analyze variables related to the primary heterotrophic growth expressed as radicle and hypocotyl elongation.

### 2.3. Dynamics of Heterotrophic Growth

Data on heterotrophic growth as a function of time were used to produce 98 graphs. These resulted from 14 graphs by accession, one for the radicle and one for the hypocotyl elongation, at seven temperatures. For illustrative purposes, only the 14 graphs of heterotrophic growth at 25 °C are presented in Figure 4. Each graph contains the plots of 30 individuals per accession.

These plots of growth at 25 °C were, in general, representative of the dynamics of radicles and hypocotyls in response to the other temperatures. The first noteworthy point is the high variability observed for radicles, regardless of their accession. Second, this variability was higher than the variability of hypocotyls. Third, radicle dynamics followed a hyperbolic path, while hypocotyls rather followed “S” shape trajectories. Fourth, the final length of radicles was slightly higher than the final length of hypocotyls (Figure 5). Indeed, the mean S:R ratios were, in general, inferior to one, with random statistical differences between temperatures (Figure 5). Ratios obtained at 40 °C showed the highest variability, possibly due to the lower number of individuals (n < 30) and the low growth observed for seedlings surviving at this temperature.

### 2.4. Embryo Axes Final Lengths

Concerning the final length of radicles and hypocotyls in response to temperature (Figure 5), they were recorded once the growth plateaus were confirmed by the measurements on at least three consecutive pictures. As for germination, the time needed to attain the final length depended on temperature treatment. Beyond the variability discussed above, the seven accessions responded in a similar manner. All these responses were perfectly fitted to parabolic functions with optimal values in the narrow range of 15 to 20 °C (Figure 5). These responses differed strongly from the responses observed for germination [22].

### 2.5. Relative Elongation Rate

Schnute’s non-linear equation was fitted to data on the time course of growth for each seedling and axis. The second derivative was used to estimate the maximum elongation rate (*MER*) and the *RER_max(T)_*. Curve fittings to data from 40 °C yielded poor results because this temperature appeared lethal to most seedlings and were not used in the growth rate analyses hereafter.

The extent of the genetic variation in maximum relative elongation rates in response to temperature, *RER_max(T)_*, was also assessed using PCA. Six components accounted for 100% of the variability. Four components explained 86.4% of the total variability contained within the 14 variables related to seedling growth that were included in the analysis (Table 2). The first two components accounted for only 53.3% of the variability.

HCPC on seedling growth rates produced the results presented in the plots of Figure 6. Accessions were positioned according to their contribution to components 1 and 2 as well as 1 and 3. In both plots, three groups could be distinguished by their location in the plot. The variety of subsp. *falcata*, ‘Krasnokutskaya’, was clearly separated from the other accessions based on PC1. ‘Flamande’, ‘Harpe’ and ‘Luzelle’ belonged to the same cluster in both plots. Further, HCPC on separated radicle and hypocotyl elongation rates under the different temperatures produced plots in which ‘Krasnokutskaya’ was positioned in a different cluster than ‘Flamande’, ‘Harpe’ and ‘Luzelle’ (Figure 7). All this reflects the different responses to temperature by these two groups. In order to confirm this, the following analyses were performed.

For each accession and axis, the *RER_max(T)_* estimated at the different temperatures were plotted together and fitted to a five-parameter Beta model (Figure 8).

Elongation rates showed high variability within temperatures for both radicle and hypocotyl elongation. Nevertheless, variability was smaller for hypocotyls than for radicles, regardless of the accession (Figure 8), suggesting that hypocotyls are less affected by temperature variations than radicles. Further, variability was the highest between 20 and 35 °C. This suggests that temperature had a significant impact on elongation rates, and the responses were not consistent across all seedlings within any accession.

Within each accession, radicle and hypocotyl elongation curves of response to temperature could be very similar, as was the case for ‘Krasnokutskaya’ (0.042 vs. 0.041 mm·h^−1^·mm^−1^) or differed significantly. For example, the maximum relative elongation rates for the radicles were faster than for hypocotyls of ‘Flamande’ (0.054 vs. 0.041 mm·h^−1^·mm^−1^), ‘Harpe’ (0.059 vs. 0.040 mm·h^−1^·mm^−1^) and ‘Luzelle’ (0.057 vs. 0.040 mm·h^−1^·mm^−1^). For these three accessions, the estimated *T_opt_* were also different for radicle and hypocotyl elongation, further highlighting divergent temperature responses.

When comparing the response curves of radicle elongations between accessions, significant differences (*p* < 0.01) were observed (Figure 8). For example, the variety ‘Harpe’ had the fastest radicle elongation rate, whereas ‘Krasnokutskaya’ had the lowest one. The response curve of the variety ‘Orca’ appeared to be unique because the *RER_max(T)_* increased linearly as temperature increased from 5 to 30 °C. The estimated *T_opt_* for *RER_max(T)_* ranged from 29.4 °C for ‘Flamande’ to 35 °C for ‘Orca’. On the other hand, some accessions had exchangeable curves.

Concerning hypocotyl elongation rates, in contrast with radicles, the response curves of the different accessions were significantly different (Figure 8). Indeed, the response curves of the variety ‘Orca’ and landrace ‘Demnate3’ appeared to be, respectively, unique. Their equations did not fit the data of any other accession properly. Furthermore, ‘Demnate3’ had the highest *RER*_*max*(32)_ (0.043 mm·h^−1^·mm^−1^), whereas ‘Orca’ had the lowest one (0.038 mm·h^−1^·mm^−1^). It is noteworthy that the hypocotyl *RER_max(T)_* of the variety ‘Orca’ increased linearly as temperature increased from 5 to 30 °C. On the other hand, the models of some accessions could fit the data of others.

### 2.6. Relationships between Relative Elongation Rates and Germination Rates

We explored the eventual relationships between maximum relative elongation rates and maximum germination rates (*α*) for the seven accessions under seven temperatures (Figure 8). As for the elongation rates, data on the germination rate were fitted to the Beta model. The response curves of accessions to temperature, in terms of *α*, were all statistically different.

For each accession, the visual inspection and comparison of elongation and germination rates curves in Figure 8 suggests that it could be difficult to establish a link between these variables. A regression analysis confirmed the weakness of such relationships (Figure 9). Overall, germination rates responded differently to temperature compared to elongation rates of the embryo axes.

## 3. Discussion

Climate change, local annual courses of precipitation and temperatures, is a significant global concern that hinders agricultural production and threatens the equilibrium of ecosystems [1]. Temperature and soil water availability are important environmental factors regulating both timing and the rates of seed germination as well as seedling establishment. Indeed, the processes of seed germination and primary heterotrophic growth are complex biological phenomena influenced by both genetic and environmental factors [23]. Additionally, primary heterotrophic growth plays a crucial role in the emergence of seedlings from the soil.

*Medicago sativa* (L.) is the most important forage legumes worldwide. The accelerated breeding of alfalfa cultivars adapted to new ranges of temperatures could be necessary under most future climate scenarios. The aim of this work was to explore the genetic diversity of responses to constant temperature during seedling heterotrophic growth and its relationship with seed germination. We examined a sample of seven *M. sativa* accessions. Seeds or seedlings were placed in the dark under eight constant temperatures in the range of 5 °C to 40 °C. Germinated seeds were manually counted, while hypocotyl and radicle growth were estimated by using image analysis and curve fitting. Multivariate analyses highlighted links between the responses and the origin of accessions.

While a considerable amount of scientific literature focuses on seed germination, the genetic diversity of responses to temperature during germination in pasture species has received limited attention [6,8,10,22]. A similar lack of emphasis is observed in the study of primary heterotrophic growth in general [24] and specifically in perennial forage crops [11].

The values of the maximum germination percentage (germinability) in response to temperature that we analyzed here are coherent with recent studies in which enough time has been given for the seed to germinate under low and high temperatures [25]. They are also coherent with the results of a recent study on 38 accessions of alfalfa from a wide range of origins [6]. It includes the specific responses of ‘Krasnokutskaya’ and other wild populations of *M. sativa* subsp. *falcata* and subsp. *sativa*, as well as some landraces such as ‘Flamande’ [6].

As expected from the theory and massive empirical evidence, the responses to temperature were non-linear. This fact is frequently disregarded, and linear relationships are erroneously proposed within narrow ranges of temperature (e.g., [26]). Furthermore, the false assumption of linearity has brought several authors to propose bilinear models for germination responses in many species [27,28,29,30,31,32,33,34].

The non-linear relationships that we quantified between temperature and germination, as well as seedling heterotrophic growth, were frequently different, as reflected by PCA analyses and curve comparisons [10]. We found accession-specific responses that could be related to the level of domestication [6,8] or the provenance of the accessions. These results could prompt physiologists to extend their analyses of responses to temperature to other processes on alfalfa [35].

Overall, variability was high, both within and between accessions, for all the response variables. Accessions showed significant differences in their non-linear response curves in terms of germinability, germination rates and seedling relative elongation rates. Nevertheless, differences were more noticeable in germinability and germination rates compared to seedling heterotrophic growth rates. Consequently, germination-related variables could be easier to use as early markers for alfalfa selection and breeding for the future.

If we consider that germination and heterotrophic growth form a continuum, then a methodological improvement could be to follow individuals from the moment seeds are placed in water until the end of their heterotrophic growth at a constant temperature.

## 4. Materials and Methods

### 4.1. Plant Material

Seven accessions of alfalfa were studied. They included ‘Krasnokutskaya’, a variety of *M. sativa* subsp. *falcata* (Syn: *M. sativa* L. nothosubsp. *varia* (Martyn) Arcang.) as well as two landraces (‘Flamande’ and ‘Demnate3’) and four varieties (‘Harpe’, ‘Luzelle’, ‘Orca’ and ‘Barmed’) of *M. sativa* subsp. *sativa* (Table 3) (http://florilege.arcad-project.org/fr/crb/especes-fourrageres, accessed on 2 September 2023). Seeds were obtained from “Centre de Ressources Génétiques des Espèces Fourragères” in Lusignan, France (INRAE-URP3F). They were stored in opaque envelopes in the dark at 5 °C and 30% relative humidity (RH) until use.

### 4.2. Germination Analysis

Unpublished data from a study on the response of alfalfa accessions to temperature during germination [22] were used to estimate germination rates (*α* in % of seeds per unit of time) and the lag time between the beginning of incubation and the beginning of germination (*t_c_* in hours). To this end, for each replicate, the cumulated number of germinated seeds was fitted to a non-rectangular hyperbole as in [6,25]. Maximum germination ratios (*G_max_*, % of germinated seeds), together with *α* and *t_c_* at the eight germination temperatures, were used to perform a principal component analysis (PCA) [6].

### 4.3. Heterotrophic Growth Measurement

Seeds were scarified between two sheets of sandpaper (grade 180) in order to break any residual physical dormancy [6]. After scarification, one thousand seeds per accession were placed for germination in the dark at constant temperature of 25 °C on top of two sheets of sterilized Whatman paper (ref. 3645 Whatman, France) in Polypropylene boxes (55 mm × 120 mm × 180 mm, GEVES trademark, Loire Plastic, France, hereafter GEVES-type box) containing 15 mL of deionized and autoclave-sterilized water. The rest of the protocol has previously been described [11]. Briefly, for each accession, two hundred and forty seedlings with a radicle longer than 1 mm were chosen, numbered and placed in groups of ten on top of blotter bleu paper (Anchor Paper, St. Paul, MN, USA) within GEVES-type boxes watered with 20 mL of deionized and autoclave-sterilised water. The groups of three boxes per accession were transferred to one of the eight temperature treatments (5 °C to 40 °C with intervals of 5 °C) in unlit growth chambers. The temperature within the useful volume of the chambers was checked by six thermocouples placed at different positions around the GEVES boxes and logged every 20 s. Further, temperature and relative humidity in the chambers were recorded every minute during the experiment.

For hypocotyl and radicle growth measurements, pictures were taken with a Nikon D70 digital camera (objective Micro Nikon 60 mm-f/2.8 in automatic mode and sensibility set to 200 ISO) at a resolution of 3008 × 2000 pixels [11]. For each one of the 30 individual seedlings by accession and temperature, at least 15 pictures were used for growth analysis. The exception was made for seedlings at 40 °C, where high and early mortality was common for most accessions. In general, these pictures were taken at variable time intervals and duration, which depended on the treatment (Table 4). ImageJ software (version 1.47, http://imagej.nih.gov/ij/, accessed on 2 September 2023) was used for image analyses via a combination of automated and manual steps.

### 4.4. Heterotrophic Growth Modelling and Calculations

The kinetics of elongation for the radicle and hypocotyl of each seedling were fitted to Schnute’s non-linear model [36] by using the least-squares method [37]. Calculations were performed as previously described [11]. Briefly, for each seedling, the first derivative of Schnute’s model was used to estimate the instantaneous absolute elongation rate (*AER*, mm·h^−1^) for each axis. *AER* was, in turn, used to calculate the relative elongation rate (*RER*, mm·h^−1^·mm^−1^) such that at time *t*,
(1)$$RER(t)=\frac{AER(t)}{y(t)},$$
where *y*(*t*) was the length of the axis (mm).

The second derivative gave the maximum elongation rate (*MER*, mm·h^−1^), which, in turn, was used to calculate the maximum relative elongation rate (*RER_max(i.j.T)_*, mm·h^−1^·mm^−1^) per axis *i* (radicle or hypocotyl) and seedling *j* (1 to 30) at temperature *T* (5 to 35 °C). For each accession and axis the *RER_max(i.j.T)_* of the 30 seedlings were pooled together and plotted against temperature. The five-parameter Beta model (Equation (2)) was fitted to each dataset, such that:(2)$${Acc}_{RER\left(T\right)}=MRAcc.\left(\left(\frac{\left(Tmax-T\right)}{\left(Tmax-Topt\right)}.\frac{\left(T-Tmin\right)}{\left(Topt-Tmin\right)}\right){\;}^{\frac{\left(Topt-Tmin\right)}{\left(Tmax-Topt\right)}}\right){{\;}^{\delta }}^{\;}$$
where *Acc_RER(T)_* is a function of temperature, *T*; *T_min_*, *T_opt_* and *T_max_* are the three cardinal temperatures; *MRAcc* is the maximum rate of the hypocotyl or radicle relative elongation estimated at *T_opt_*, and *δ* is a shape parameter [38]. These boundaries were fixed at 0 °C, under the assumption that no vascular plant can grow below 0 °C [39], and our observations of growth failure at 40 °C.

Shoot-to-root (S:R) ratios were calculated from the length of the hypocotyl and radicle in the last image used for the curve fitting of each seedling. All calculations were performed as previously described [11].

### 4.5. Statistical Analyses

In addition to the curve fittings described before and exploratory data analyses, PCA and hierarchical clustering on principal components (HCPC) [40] were performed on the seven accessions and described by 24 variables related to germination. To this end, for each accession, the means of the four replicates per temperature were calculated for variables *G_max_* and estimated *α* and *t_c_*, such that *G_max_T_*, *α_T* and *t_c_T_*, with *T,* equaled 5 to 40 by steps of 5 (Table 4) This gave the 24 response variables for multivariate analyses.

Correlation analyses were performed between germination and growth rates. All these analyses were performed with R (R software version 4.1.3, https://www.r-project.org, (accessed on 10 March 2022), packages: base, stats, FactoMaineR and nlstools- R Core Team, 2022 [41]).

## 5. Conclusions

Accelerated breeding of alfalfa cultivars adapted to new ranges of temperature could be necessary under most future climate scenarios. This work aimed to explore the genetic diversity of a sample of accessions for the responses to temperature during seed germination and seedling heterotrophic growth. Seeds or seedlings were placed in the dark under eight constant temperatures in the range of 5 °C to 40 °C. Germinated seeds were manually counted, while hypocotyl and radicle growth were estimated by image analysis and curve fittings. Multivariate analyses highlighted links between the responses and the origin of accessions. Variability was high, both within and between accessions, for all the response variables. Accessions showed significant differences in their non-linear response curves in terms of germinability, germination rates and relative seedling elongation rates. Nevertheless, differences were more noticeable in germination variables compared to seedling heterotrophic growth. Consequently, germinability and germination rates could be easier to use early markers for alfalfa selection and breeding for the future.

## Figures and Tables

**Figure 1 plants-12-03224-f001:**
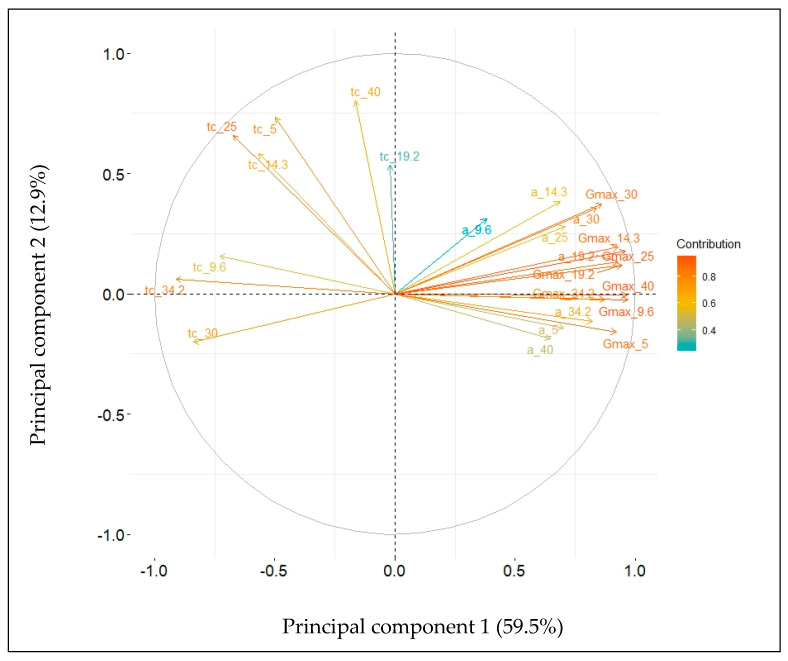
PCA on 24 variables related to seed germination under eight constant temperatures for seven accessions of alfalfa (*Medicago sativa* L.). Normalized contribution of the variables to components 1 and 2.

**Figure 2 plants-12-03224-f002:**
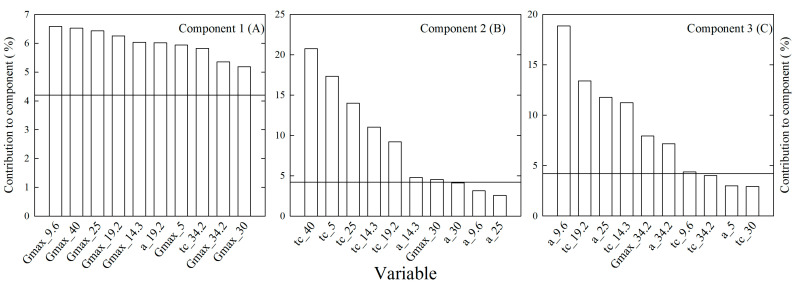
Ten major seed germination-related variables contributing to component 1 (**A**), 2 (**B**) or 3 (**C**) of the PCA of the responses to eight constant temperatures by seven accessions of alfalfa (*Medicago sativa* L.).

**Figure 3 plants-12-03224-f003:**
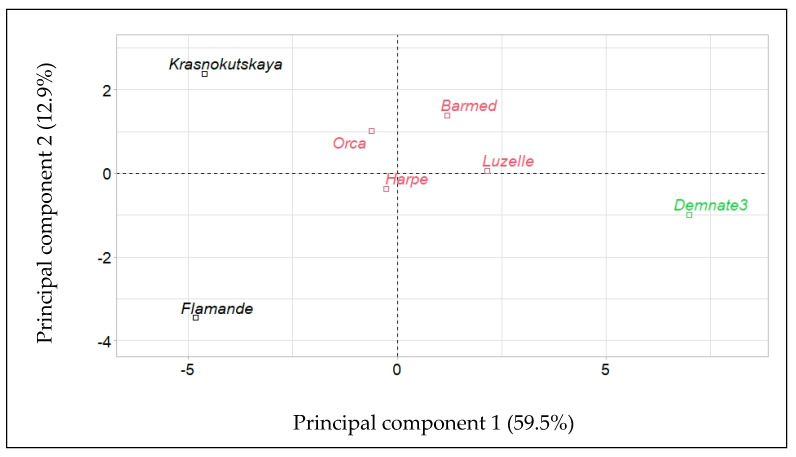
Hierarchical clustering into three groups (represented by colors) of the seven accessions of alfalfa (*Medicago sativa* L.) tested for seed germination under eight constant temperatures.

**Figure 4 plants-12-03224-f004:**
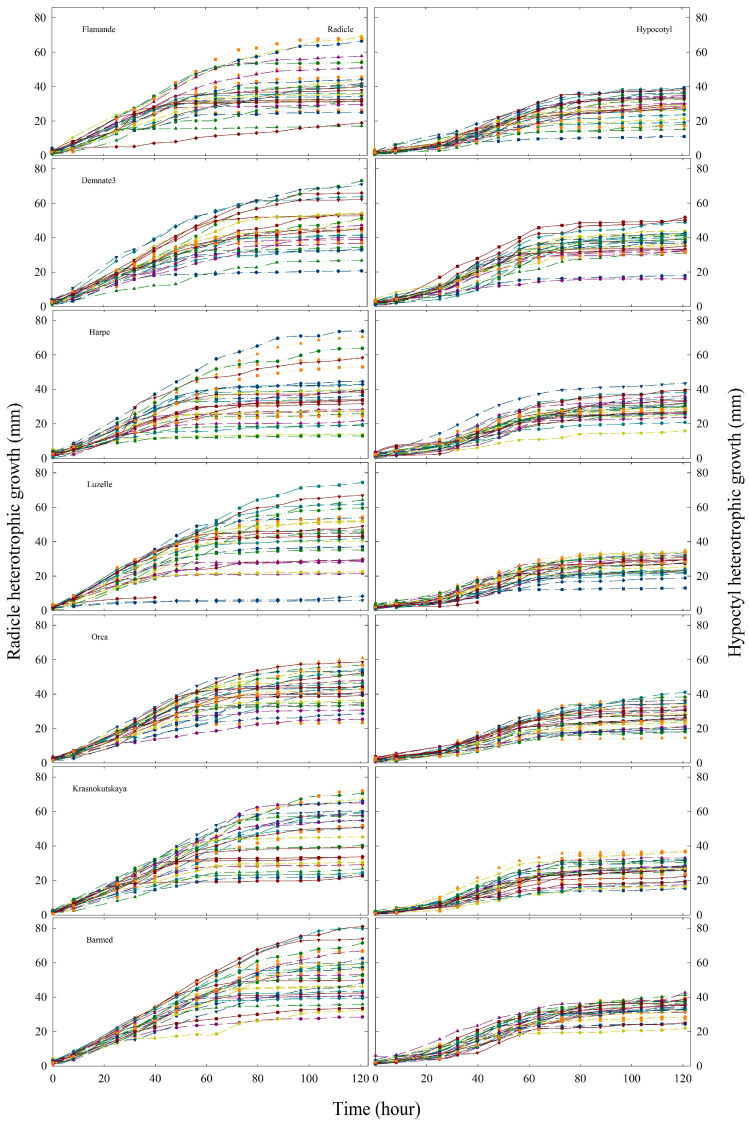
Time courses of the heterotrophic growth at 25 °C temperature observed for seven accessions of (*Medicago sativa* L.). n = 30.

**Figure 5 plants-12-03224-f005:**
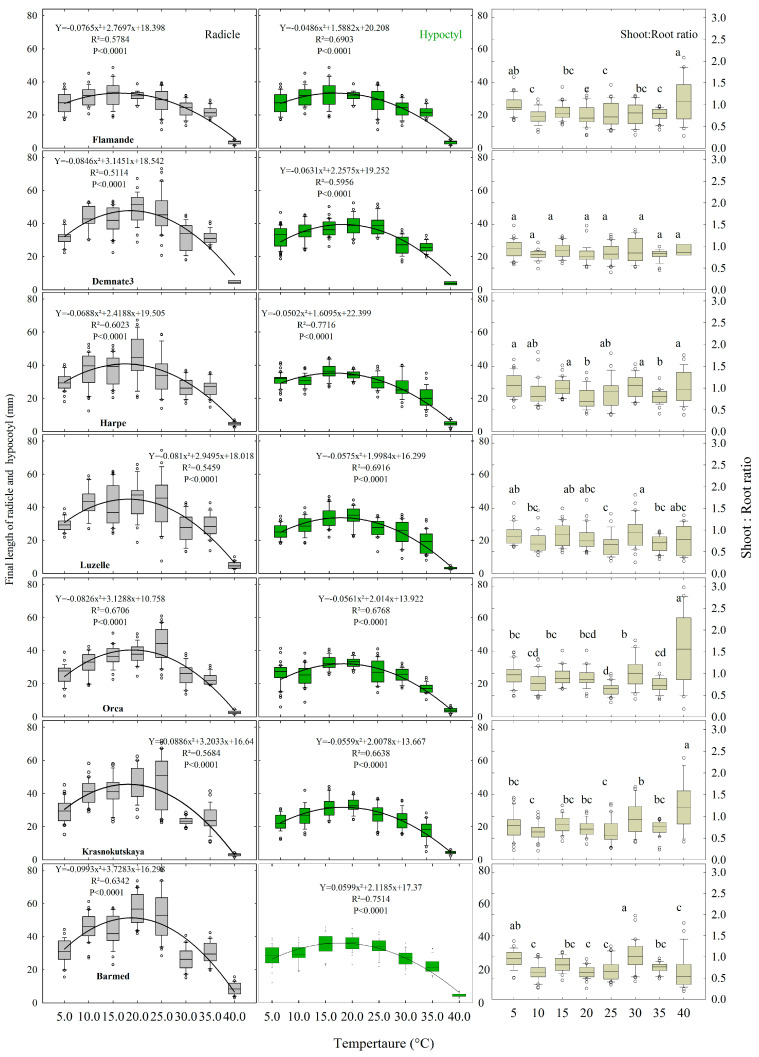
Box plot for the final length of radicle and hypocotyl (**left** and **center**) and shoot:root ratio (**right**) for the seven accessions of (*Medicago sativa* L.) studied for their response to constant temperature during heterotrophic growth. For each temperature between 5 and 34.2 °C, n = 30. For 40 °C, 0 ≤ n ≤ 30 because of seedling mortality. Within each plot, boxes with the same letter are not significantly different.

**Figure 6 plants-12-03224-f006:**
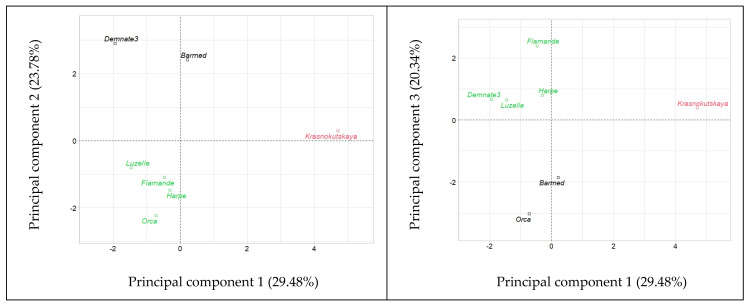
Hierarchical clustering into three groups (represented by colors) of the seven accessions of alfalfa (*Medicago sativa* L.) tested for combined heterotrophic growth of radicle and hypocotyl under seven constant temperatures.

**Figure 7 plants-12-03224-f007:**
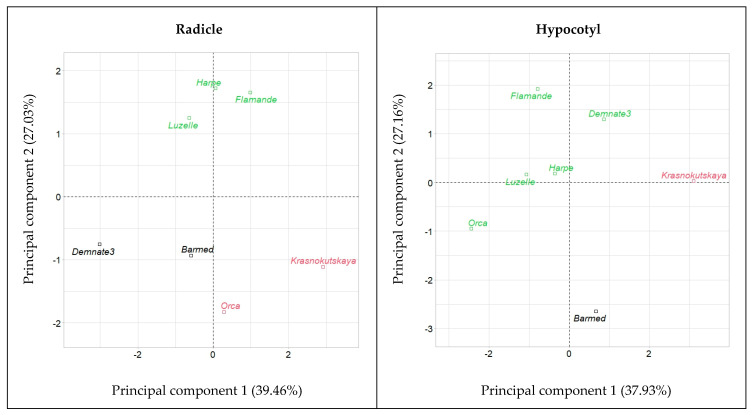
Hierarchical clustering into three groups (represented by colors) of the seven accessions of alfalfa (*Medicago sativa* L.) tested for heterotrophic growth under seven constant temperatures.

**Figure 8 plants-12-03224-f008:**
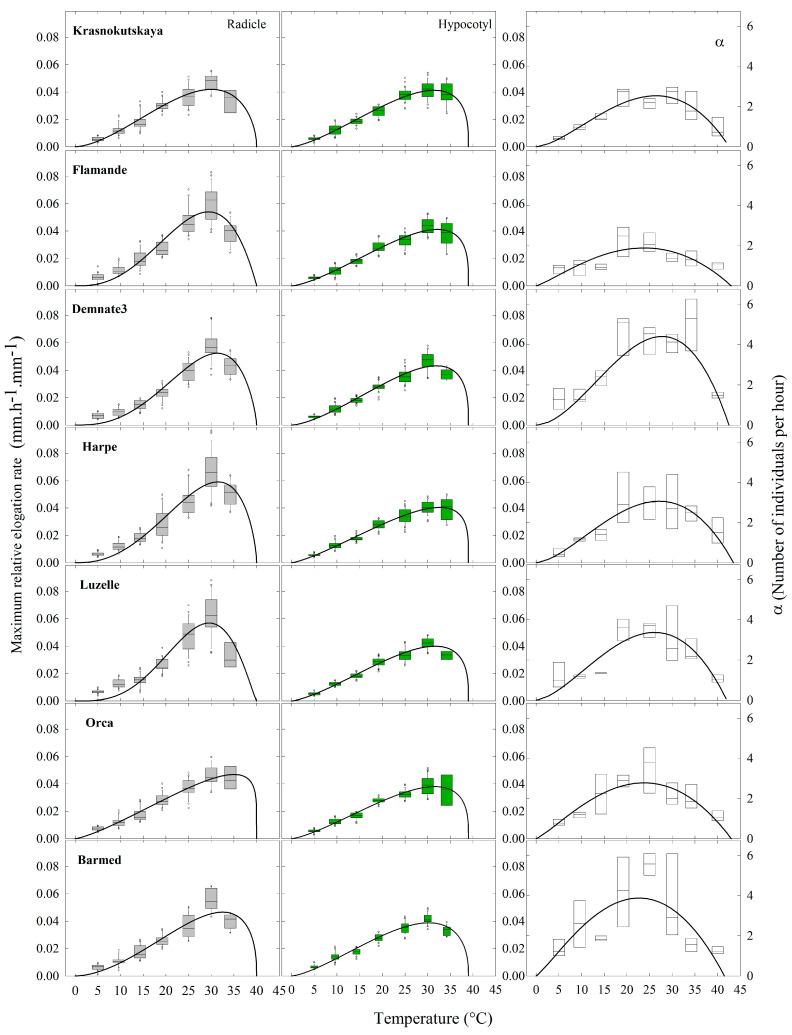
Box plot for maximum relative elongation rate of radicle and hypocotyl (**left** and **center**) and maximum germination rate, *α*, (**right** colon) observed for seven accessions of (*Medicago sativa* L.) during heterotrophic growth in response to constant temperatures. All box plots were fitted with the Beta function. For each temperature between 5 and 34.2 °C, n = 30. For 40 °C, 0 ≤ n ≤ 30 because of seedling mortality.

**Figure 9 plants-12-03224-f009:**
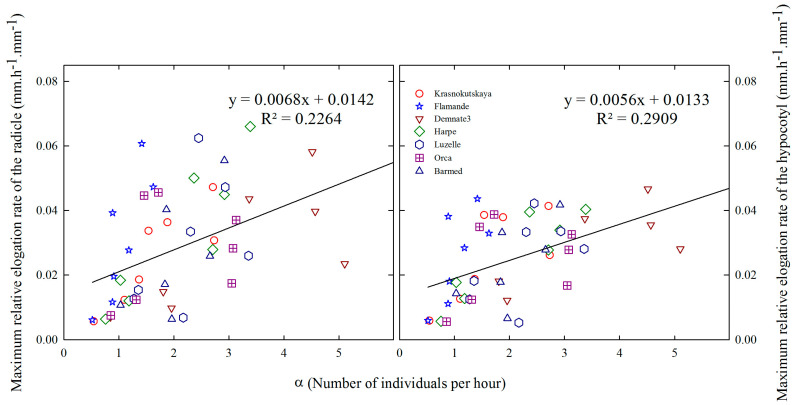
Relationship between maximum seed germination rate (*α*) and maximum relative elongation rate of radicle (**left**) and hypocotyl (**right**) during heterotrophic growth for seven accessions of (*Medicago sativa* L.). For elongation rates, each point is the mean n = 30 for temperatures between 5 and 34.2 °C.

**Table 1 plants-12-03224-t001:** The first four components, eigenvalues (latent roots) of the correlation matrix, the percentage of total variability accounted by each component and the cumulative percentage from the PCA on germination responses to temperature for seven accessions of *Medicago sativa* L.

Component	Eigenvalue	Percentage of Variability	
		Component	Cumulative
1	14.288	59.5	59.5
2	3.099	12.9	72.4
3	2.882	12.0	84.4
4	1.744	7.3	91.7

**Table 2 plants-12-03224-t002:** The first four components, eigenvalues (latent roots) of the correlation matrix, the percentage of total variability accounted for by each component and the cumulative percentage from the PCA on heterotrophic growth responses to temperature for seven accessions of *Medicago sativa* L.

Component	Eigenvalue	Percentage of Variability	
		Component	Cumulative
1	4.127	29.48	29.48
2	3.329	23.78	53.26
3	2.848	20.34	73.61
4	1.795	12.82	86.43

**Table 3 plants-12-03224-t003:** Identification of accessions of *Medicago sativa* L. used in this investigation (http://florilege.arcad-project.org/fr/crb/especes-fourrageres, accessed on 2 September 2023.

Accession	Taxon	Autumn Dormancy	Type	Collection Site	Latitude and Longitude	Altitude above Sea Level (m)	Mean Temperature Warmest Quarter (°C)	Mean Temperature Coldest Quarter (°C)	Precipitation Warmest Quarter (mm)	Precipitation Coldest Quarter (mm)
Flamande	*M. sativa* spp. *sativa* L. (4×)	4	Landrace	Calais, France	50°57′04.64″ N, 1°51′31.27″ E	27	16.6	3.3	181	154
Demnate3	*Idem*	9	Landrace	Demnate, Marocco	31°43′55.51″ N, 7°00′41.25″ W	923	24.7	9.4	17	154
Harpe	*Idem*	4	Variety							
Luzelle	*Idem*	3	Variety							
Orca	*Idem*	4.5	Variety							
Barmed	*Idem*	7	Variety							
Krasnokutskaya	*M. sativa* spp. *falcata* (L.) Arcang ^1^ (4×)		Variety	Russia						

^1^ Syn: *M. sativa* L. nothosubsp. *varia* (Martyn) Arcang. The prefix “notho” indicates that the taxon originated through hybridization.

**Table 4 plants-12-03224-t004:** Temperatures, relative humidity and vapor pressure deficit (VPD) measured for heterotrophic growth.

Temperature Treatment (°C)	Actual Temperature (°C)	Relative Humidity (%)	VPD (kPa)	Sampling Frequency (h)	Sampling Period (h)	Equivalent Growing Degree-Day
5	5.0	74	0.23	48	672	140
10	9.6	84	0.20	16	224	90
15	14.3	76	0.41	12	168	100
20	19.2	74	0.61	8	112	90
25	25.0	80	0.63	8	112	117
30	30.0	50	2.12	8	112	140
35	34.2	65	1.97	12	168	239
40	40.0	55–100	3.3–0.0	168	1344	2240

## Data Availability

Data are available from the authors.

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
