# Peer review of "Early Detection of Phenotypic Diversity of Alfalfa (*Medicago sativa* L.) in Response to Temperature"

_plants, 2023, doi:10.3390/plants12183224_

Round 1

Reviewer 1 Report

Minor corrections need to be addressed in the manuscript:

(1) Present the important characteristics of the seven genotypes used in the study, as a part of Table 3

(2) The authors need to present an elaborate discussion in the manuscript, which is otherwise very minimally presented now

(3) There is lack of clarity on number of replications used in the study

Author Response

Thank you very much for taking the time to review our manuscript and for your feedback. Please find the detailed responses below and the corresponding revisions/corrections highlighted/in track changes in the re-submitted files.

Point-by-point response to Comments and Suggestions for Authors

Comments 1: Present the important characteristics of the seven genotypes used in the study, as a part of Table 3

Response 1: Thank you for suggesting this out.  We already included in this table the information pertinent to the scope of our work. We disagree with the need to expand the table that should make it cumbersome.

Comments 2: The authors need to present an elaborate discussion in the manuscript, which is otherwise very minimally presented now

Response 2: Thank you for your comment.  We have restructured and expanded the discussion keeping it concise and focus on the aim and scope of the manuscript.

Comments 3: There is lack of clarity on number of replications used in the study

Response 3: We have check out carefully.  The number of replicates is explicit in the M&M section as well as in each figure.  We could not see another place to better clarify this point.

Reviewer 2 Report

Review

The relevance of the work is beyond doubt. Evaluation of alfalfa varieties for temperature regime at germination is important information for breeders and practitioners. However, there are remarks on the methodology presentation, the elimination of which would improve the article.

1.In section 4. Materials and methods (line 304),

4.1 Plant material (line 305) it is recommended to specify the year of research.

If I understand correctly, the article submitted for publication is a further processing of experimental material obtained in 2019 and partially published by the authors in the article they refer to https://doi.org/10.15258/sst.2019.47.3.10 Or is this a new study?

2. The expression "The wild population of subsp. falcata, 'Krasnokust-(line 186) kaya'" is not correct. The variety 'Krasnokustkaya' (M. sativa L. nothosubsp. varia (Martyn) Arcang.) is an official variety obtained in 1973 and entered in the State Variety Register of Russia. It is not a wild population.

3. In the conclusions it is written:

«Multivariate analyses highlighted links between responses and the 403

origin of accessions. Variability was high, within and between accessions, for all the 404

response variables ..405»

In my opinion, it would be helpful for the readers of the article to see the names of the specific forms and accessions for which these data were obtained, and more specific recommendations for the varieties studied.

Author Response

1. Summary
Thank you very much for taking the time to review our manuscript and for your feedback. Please find the detailed responses below and the corresponding revisions/corrections highlighted/in track changes in the re-submitted files.

2. Point-by-point response to Comments and Suggestions for Authors 

Comments 1: In section 4. Materials and methods (line 304), 
4.1 Plant material (line 305) it is recommended to specify the year of research.  If I understand correctly, the article submitted for publication is a further processing of experimental material obtained in 2019 and partially published by the authors in the article they refer to https://doi.org/10.15258/sst.2019.47.3.10 Or is this a new study?

Response 1:  Yes, you understood correctly.  As said in the manuscript, we analyzed unpublished data of germination recorded for the 2019 paper.  We used the same seedlots for the heterotrophic growth study performed afterwards. 

Comments 2: The expression "The wild population of subsp. falcata, 'Krasnokust-(line 186) kaya'" is not correct. The variety 'Krasnokutskaya' (M. sativa L. nothosubsp. varia (Martyn) Arcang.) is an official variety obtained in 1973 and entered in the State Variety Register of Russia. It is not a wild population. 

Response 2: Thank you for your very informative comment.  We have corrected the whole manuscript accordingly.  We included a note to express that, taxonomically: M. sativa subsp. falcata is sometimes named M. sativa L. nothosubsp. varia (Martyn) Arcang.) in order to indicate that the taxon originated through hybridization.

Comments 3: In the conclusions it is written:
«Multivariate analyses highlighted links between responses and the (403) origin of accessions. Variability was high, within and between accessions, for all the (404) response variables. (405)»
In my opinion, it would be helpful for the readers of the article to see the names of the specific forms and accessions for which these data were obtained, and more specific recommendations for the varieties studied.

Response 3: We have check out carefully the Results section.  We found the information suggested by the reviewer to be included in the conclusion.  We think that the conclusion section is not, in our case, the place to talk about specific points already presented elsewhere in the manuscript.